# Large Language Models as General Pattern Machines

**Suvir Mirchandani[1], Fei Xia[2], Pete Florence[2], Brian Ichter[2], Danny Driess[2 3],**
**Montserrat Gonzalez Arenas[2], Kanishka Rao[2], Dorsa Sadigh[1 2], Andy Zeng[2]**
[1]Stanford University, [2]Google DeepMind, [3]TU Berlin
https://general-pattern-machines.github.io

**Abstract:** We observe that pre-trained large language models (LLMs) are capable of autoregressively completing complex token sequences—from arbitrary ones procedurally generated by probabilistic context-free grammars (PCFG), to more rich spatial patterns found in the Abstraction and Reasoning Corpus (ARC), a general AI benchmark, prompted in the style of ASCII art. Surprisingly, pattern completion proficiency can be partially retained even when the sequences are expressed using tokens randomly sampled from the vocabulary. These results suggest that without any additional training, LLMs can serve as general sequence modelers, driven by in-context learning. In this work, we investigate how these zero-shot capabilities may be applied to problems in robotics—from extrapolating sequences of numbers that represent states over time to complete simple motions, to least-to-most prompting of reward-conditioned trajectories that can discover and represent closed-loop policies (e.g., a stabilizing controller for CartPole). While difficult to deploy today for real systems due to latency, context size limitations, and compute costs, the approach of using LLMs to drive low-level control may provide an exciting glimpse into how the patterns among words could be transferred to actions.

**Keywords:** large language models, in-context learning, language for robotics

## 1 Introduction

Large language models (LLMs) are trained to absorb the myriad of patterns that are woven into the structure of language. They not only exhibit various out-of-the-box capabilities such as generating chains of reasoning [1, 2], solving logic problems [3, 4], and completing math puzzles [5], but also have been applied in robotics where they can serve as high-level planners for instruction following tasks [6, 7, 8, 9, 10, 11, 12], synthesize programs representing robot policies [13, 14], design reward functions [15, 16], and generalize user preferences [17]. These settings rely on few-shot in-context examples in text prompts that specify the domain and input-output format for their tasks [18, 19], and remain highly semantic in their inputs and outputs.

A key observation of our work—and perhaps contrary to the predominant intuition—is that an LLM's ability to represent, manipulate, and extrapolate *more abstract, nonlinguistic* patterns may allow them to serve as basic versions of *general pattern machines*. To illustrate this idea, consider the Abstraction and Reasoning Corpus [20], a general AI benchmark that contains collections of 2D grids with patterns that evoke abstract concepts (e.g., infilling, counting, and rotating shapes). Each problem provides a small number of input-output examples, followed by test input(s) for which the objective is to predict the corresponding output. Most methods (based on program synthesis) are manually engineered with domain-specific languages [21, 22, 23, 24] or evaluated on simplified extensions or subsets of the benchmark [25, 26, 27]. End-to-end machine learning methods only solve a handful of test problems [28]; however, our experiments indicate that LLMs in-context prompted in the style of ASCII art (see Fig. 1) can correctly predict solutions for up to 85

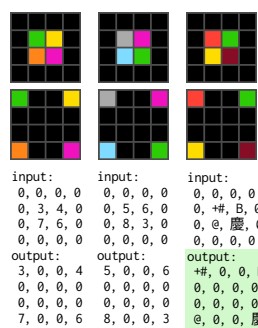

Fig. 1: LLMs out-of-the-box can complete (highlighted) complex ARC patterns [20] expressed in arbitrary tokens.

(out of 800) problems—exceeding some existing recent systems [21, 22, 24], without additional model training or fine-tuning. Surprisingly, we find this extends beyond ASCII numbers, and that when they

7th Conference on Robot Learning (CoRL 2023), Atlanta, USA.

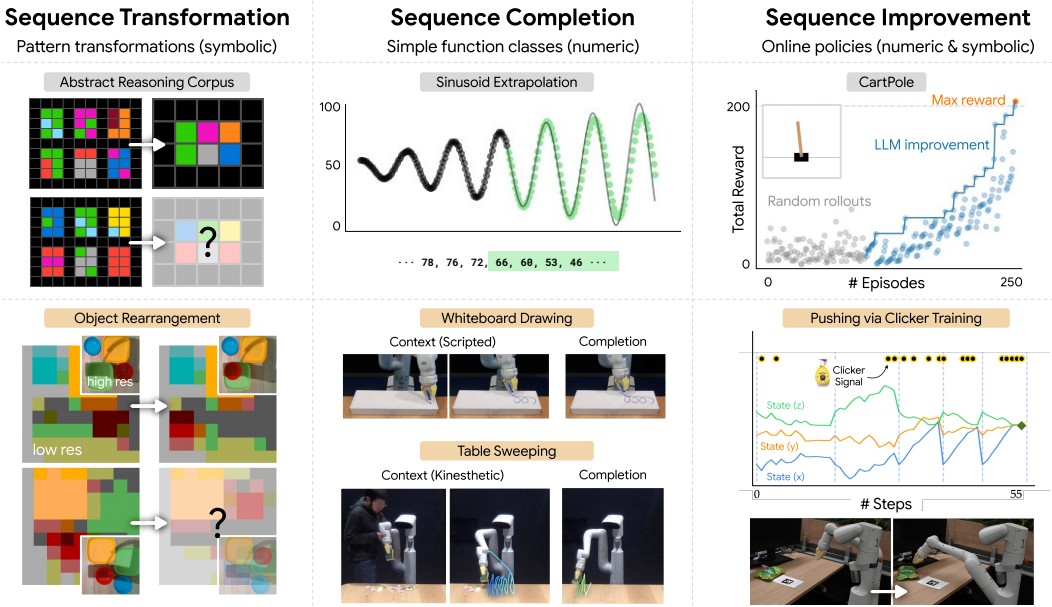

Fig. 2: Pre-trained LLMs out-of-the-box may serve as basic versions of *general pattern machines* that can recognize and complete sequences of numeric or arbitrary (symbolic) tokens expressing abstract problems in robotics and sequential decision-making. Experiments show that to an extent, LLMs can in-context learn (i) sequence transformations (e.g., to reason over spatial rearrangements of symbols, for dynamics modeling and next state prediction on downsampled images), (ii) completion of simple functions (e.g., to extrapolate kinesthetic demonstrations), or (iii) meta-patterns to improve return-conditioned policies (e.g., to discover oscillatory behaviors to stabilize a CartPole).

are replaced with a mapping to *randomly sampled tokens* in the vocabulary, LLMs can still generate some valid solutions. These results suggest an intriguing insight: that LLMs may exhibit more general capabilities of representing and extrapolating symbolic patterns, invariant to the specific tokens involved. This is in-line with—and complementary to—recent observations that using random or abstract label mappings for in-context classification retains some performance compared to ground-truth labels [29, 30]. We hypothesize that the capabilities that drive pattern reasoning on the ARC may allow general pattern manipulation at various levels of abstraction useful for robotics and sequential decision making [31, 32], wherein a diverse array of problems involve patterns that may be difficult to reason about precisely in words. For example, a procedure for spatially rearranging tabletop objects could be represented using arbitrary tokens (see Fig. 2). As another example, optimizing a trajectory with respect to a reward function can be framed as extrapolating a sequence consisting of state and action tokens with increasing returns.

Orthogonal and complementary to efforts that develop multi-task policies by pre-training on large amounts of robot data [33], or robotics foundation models [34] that can be fine-tuned for downstream tasks [35, 36, 37], our goal is instead to (i) assess the zero-shot capabilities that LLMs may already contain to perform some degree of general pattern manipulation, and (ii) investigate how these abilities can be used in robotics. These capabilities are certainly *not* sufficient to replace specialized algorithms; nonetheless, they are useful to characterize, and doing so may help inform priorities for training generalist models in robotics.

We assess LLMs as pattern machines categorized into three areas: sequence transformation, sequence completion, and sequence improvement (Fig. 2). First, we show that LLMs are capable of generalizing certain sequence transformations of increasing complexity with a degree of token invariance, and posit that this can carry over to spatial reasoning capabilities in robotic tasks. Next, we assess LLMs' ability to complete patterns from simple functions (e.g., sinusoids) and show this can be applied to robotic tasks like extending a wiping motion from kinesthetic demonstrations, or drawing patterns on a whiteboard. The combination of in-context sequence transformation and extrapolation further enables LLMs to do basic forms of sequence improvement. We show that providing reward-labeled trajectories as context, coupled with online interaction, can enable an LLM-based agent to learn to navigate through a small grid, discover a simple CartPole controller, and optimize simple trajectories via human-in-the-loop "clicker" reward training. Code, benchmarks, and videos are made available at https://general-pattern-machines.github.io.

## 2 Related Work

**In-Context Learning.** Pattern reasoning by prompting pre-trained LLMs with few-shot input-output examples is driven by in-context learning [38, 39]. The examples serve as a form of task specification, where the model is expected to complete further instances of the task by predicting what comes next. In-context learning extends the concept of "task prefixes" (predefined token sequences, e.g., [40]), but swapped in with examples instead. Brown et al. [39] observe that it improves (in particular, out-of-distribution generalization) from scaling model size. This is in contrast to scaling models for pre-training + fine-tuning, which has been shown to not necessarily improve OOD generalization on language tasks [41]. Nonetheless, despite compelling OOD generalization abilities, in-context learning still comes at a cost, as it continues to lag behind in terms of absolute performance on benchmarks compared to task-specific fine-tuning [38, 42].

**Explanations of In-Context Learning.** In-context learning is explicitly trained for by packing examples from the same task and dataset into a context buffer that is fed as input to an LLM with an unsupervised autoregressive objective [39], sometimes referred to as meta-training. However, it can also emerge implicitly from training on datasets where tokens exhibit a Zipfian distribution [43] on Transformer architectures, but not necessarily with recurrent architectures (e.g., vanilla RNNs) [43]. Other works have shown that in-context learning with Transformers can learn simple function classes on par with least squares [44, 45, 46], and can generalize to a seemingly unbounded number of tasks (when trained on tasks from the same task family) better than multitask MLPs [47], with Bayesian interpretations of this phenomenon [48] [49].

**In-Context vs. In-Weights Learning.** In-context learning occurs during inference without gradient updates to the model weights, and can be differentiated from in-weights learning, which relies on information stored in the model weights during LLM training [50] (and can be useful for completion tasks such as "Abraham Lincoln was born in ___"). Chan et al. [50] observes that generalization of in-context learning can be characterized as more "exemplar-based" (on the basis of similarity to in-context examples [51]), as opposed to generalization of in-weights learning which tends to be more "rule-based" (on the basis of minimal features that support category boundaries in the training data [52]). The vast capabilities of LLMs [39, 53, 54, 55, 56] have been driven by a combination of both forms of learning. In this work, we are particularly interested in in-context learning, and (depending on the task) using the semantic priors of numeric tokens to drive capabilities such as sequence completion (Section 5) and improvement (Section 6).

**LLMs and Robotics.** LLMs have been applied across several areas in robotics—such as decomposing high-level task descriptions to mid-level plans [6, 7, 57, 58, 59, 60], robot code [13, 17, 14, 61], and planning domain definition languages [10]. These methods leverage semantic priors stored in LLMs to compose plans or parameterize primitive APIs, but whether LLMs can directly influence control (e.g., at the level of trajectories) in a zero-shot manner remains an open problem. We explore how pattern reasoning capabilities of LLMs may drive various control tasks, to extend or optimize low-level sequences. While it is possible to explicitly train models for these capabilities [62, 63, 64, 65], this work focuses on the inherent abilities of LLMs out-of-the-box, which may have implications for the role of language pre-training for building embodied AI systems. Related to our work are [42] which studies how LLMs perform on non-language classification and regression tasks; [66] which examines analogical reasoning in various text tasks; and [67] which studies how LLMs can represent a rollout policy and world model in-context and then uses Q-learning to drive policy improvement across a collection of toy environments with linguistic representations. Our use of LLMs for sequence improvement can be seen as a simplification of in-context policy iteration that supports learning from demonstrations and in-context RL, driven by the generality of LLMs as pattern machines.

## 3 Language Models as General Pattern Machines

The capacity of LLMs to act as general pattern machines is driven by their ability to perform in-context learning on sequences of numeric or arbitrary tokens. An LLM typically represents sequence modeling autoregressively, with a decoder-only Transformer [68], by factorizing the probability of a sequence $x$, which is a sequence of symbols $(s_1, ..., s_n)$, into the product of conditional probabilities $p(x) = \prod_{i=1}^{n} p(s_i | s_1, ..., s_{i-1})$. To perform in-context learning, the model can be conditioned with a prompt that provides the initial tokens in the sequence $s_{1:k} = (s_1, ..., s_k)$ and uses the model to complete $s_{k+1:n}$.

The adaptability of in-context learning lies in the amount of flexibility that can be packed into $s_{1:k}$—this prompt sequence can itself contain many sequences, each an input-output pair, and perhaps additional task conditioning [38, 29]. Specifically, a model can in-context learn to complete a prompt which is a set of $N$ examples $s_{1:k} = (x^1, x^2, ..., x^N)$ where each $x^i$ is a variable-length sequence $(s_1^i, s_2^i, ..., s_{m^i}^i)$.

Rather than investigating in-context learning with natural language tasks [39], in this work we are interested in investigating more abstract notions of non-linguistic patterns. The following sections evaluate these capabilities across LLMs, and show how they can be used in robotics. By varying the notion of what each $x^i$ should be, we can characterize in-context pattern learning capabilities into the following 3 categories.

- **Sequence Transformation** (Section 4): each $x^1, ..., x^{N-1}$ is a sequence-to-sequence input-output pair; i.e., $x^i = (x_{\text{input}}^i, x_{\text{output}}^i)$, each subsequence of variable length, and $x^N$ is the query input $(x_{\text{input}}^N)$.

- **Sequence Completion** (Section 5): rather than containing input-output pairs, and rather than containing many examples of different sequences, the prompt $x = (s_1, ..., s_k)$ corresponds to discrete samples from a single function, e.g., of the form $s_i = a \cdot \sin(bi)$, which can be extrapolated.

- **Sequence Improvement** (Section 6): each $x^1, ..., x^{N-1}$ is a collection of trajectories (potentially labeled with corresponding total rewards), and $x^N$ prompts the model to "improve" the sequences by inferring a better one, e.g., with least-to-most prompting [69]—this process can be iterative and applied to a variety of formulations, e.g., offline trajectory optimization or online in-context reinforcement learning.

## 4 Sequence Transformation

LLMs are capable of in-context learning the distribution of functions that represent sequence transformations by completing abstract patterns observed among examples of input-output sequences $x^i = (x_{\text{input}}^i, x_{\text{output}}^i)$ of arbitrary tokens, each drawn from a fixed alphabet $\mathcal{A}$. For example, suppose that we are given a string of input-output examples such as "5 3 0, 3 5; 7 6 1, 6 7; 9 2 3, 2 9; 4 8 5,". Here $\mathcal{A}$ consists of tokens that represent space-prefixed digits 0–9, a comma token to separate inputs from outputs, and a semi-colon token to delineate examples from each other. A general pattern machine should infer the completion "8 4" by recognizing that the pattern is to swap the first 2 tokens, then remove the 3rd.

We use the ARC [20] to evaluate LLMs on such sequence transformations that are substantially more complex, covering a range of abstract spatial tasks: infilling, counting, rotating shapes, etc. Each task has input-output examples (3.3 on average), and 1-3 test inputs which can be represented as 2D grids. Input and output sizes may differ. LLMs can be used for the ARC by flattening grids and predicting output grid items in row-major order, which naturally supports variable-length outputs. While LLMs are not specifically trained for rasterizing spatial outputs, we hypothesize that a general pattern machine would be capable of implicitly recognizing long-range dependencies between rows (using positional encoding as a bias [71]) to pick up patterns that extend across the 2nd dimension.

| Method | Total (of 800) |
|---|---|
| (g4) gpt-4-0613 | 77 |
| (d3) text-davinci-003 | 85 |
| (d3) w/ random $\mathcal{A}$ | [†]44±6 |
| (d2) text-davinci-002 [53] | 64 |
| (p) PaLM [55, 56] | 42 |
| (d1) text-davinci-001 [39] | 11 |
| (d1) finetuned | 9 |
| Ainooson et al., 2023 [23] | [*]130 |
| Kaggle 1st Place, 2022 [70] | [#]164 |
| Xu et al., 2022 [22] | [††]57 |
| Alford et al., 2021 [24] | [**]22 |
| Ferré et al., 2021 [21] | 32 |

[†]Numbers averaged across 5 randomly sampled alphabets.
[*]Based on brute force search over a hand-designed DSL.
[#]Reported out of 400 train tasks, among 3 candidates.
[††]Reported out of a subset of 160 object-oriented problems.
[**]Based on program synthesis, out of 36 symmetry tasks.

Tab. 1: LLMs out-of-the-box can solve a non-trivial number of ARC problems.

**Result: ARC benchmark.** Table 1 shows that LLMs (PaLM, InstructGPT series in acronyms d1 - d3) prompted with input grids represented as tokens drawn from an alphabet of digits, can correctly infer solutions for up to 85 problems. Surprisingly, this outperforms some recent systems [21, 22] based on program synthesis that use manually engineered domain-specific languages (DSLs). While LLMs have yet to surpass brute-force search [23] to compose functions from a handcrafted API of grid operators, they do exhibit non-trivial performance. (We address the important caveat that parts of the ARC may be present in the training data of LLMs later below.) Note that while we are concerned with LLM performance over raw patterns, concurrent work finds improvements via object representations [72] and hypothesis search [73].

**Observation: consistent tokenization matters.** The ARC can be found among the suite of tasks in BIG-Bench [74], but has often been overlooked since many language models appear to perform poorly (near or

at zero performance). We observe this occurs due to the formatting of the benchmark, where grid elements are represented as neighboring characters i.e., "8686" (instead of " 8 6 8 6"). While subtle, this difference is enough for certain Byte-Pair Encoding (or SentencePiece) tokenizers [75, 76] (that do not tokenize per digit) to group multiple grid elements ("8" and "6") into a single token ("86") which maps to a different token embedding. This causes inconsistencies with how patterns are expressed at the token level. For example, given a task expressed as "8686, 6868; 7979," if the tokenizer groups pairs of digits 86, 68, 79, respectively, the sequential inductive patterns of the task (to swap and repeat individual digits) are lost. A simple work-around is to directly pass token indices or embeddings to the language model, or use token alphabets unlikely to be grouped together (which involves some knowledge about the tokenizer). Even beyond the ARC, we observe it is beneficial to tokenize consistently with the pattern being represented.

**Observation: token mapping invariance.** The hypothesis that LLMs can serve as general pattern machines stems from the observation that they can still solve a non-trivial number of ARC problems using alphabets $\mathcal{A}$ sampled randomly from the LLM's token vocabulary. For instance, given a particular alphabet: { 8 $\mapsto$ falls, 6 $\mapsto$ +#, 7 $\mapsto$ Ul, 9 $\mapsto$ Chev, 3 $\mapsto$ 慶, 2 $\mapsto$ 2010}, a pattern machine at sufficient proficiency can be expected to complete the prompt "falls +# falls +#, +# falls +# falls; Ul Chev Ul Chev, Chev Ul Chev Ul; 慶 2010 慶 2010," by predicting " 2010 慶 2010 慶". For example, text-davinci-003 [53, 39] with the following mapping $\mathcal{A}=\{$ 0 $\mapsto$ offence, 1 $\mapsto$ Subject, 2 $\mapsto$ Lub, 3 $\mapsto$ Fail, 4 $\mapsto$ Chev, 5 $\mapsto$ symb, 6 $\mapsto$ swung, 7 $\mapsto$ Ul, 8 $\mapsto$ escalate, 9 $\mapsto$ Chromebook} solves 52 ARC problems, and across 5 random alphabets solves an average of 43.6 problems. Interestingly, we find that token mapping invariance holds to an extent on patterns over randomly sampled *embeddings* as well (not associated with any token in the vocabulary; see Appendix A.3).

The implications of token mapping invariance are two-fold. First, note that it is possible that parts of the ARC are present in the LLM's training data (i.e., due to contamination). Thus, measuring the performance of LLMs under random alphabets may provide a closer estimate of their underlying sequence transformation capabilities. (As further evidence that these abilities are not simply due to memorization, we provide a new procedurally-generated pattern transformation benchmark described below.) Second, we hypothesize that the pattern manipulation capabilities implied by token invariance could help drive positive transfer from patterns learned across Internet-scale language data to new modalities or symbolic representations for robot reasoning. As an example, (i) Fig. 10 (top) in the Appendix shows a grasp (Skittles) detector which outputs target coordinates within a downsampled image (with 6 in-context examples), and (ii) Fig. 10 (bottom) shows spatial rearrangement via predicting simple forward dynamics where the red bowl moves to the green plate (with 9 in-context examples of downsampled images as inputs and outputs). The generality of what the arbitrary tokens could represent may allow pattern transformation capabilities—especially as LLMs improve—to be leveraged at various levels of abstraction in robotics (e.g., pixels or joint positions).

**Result: PCFG benchmark.** The ARC is a difficult benchmark, and the performance falloff can be steep (and relatively uninformative) across LLMs with decreasing model size and data scale, making it difficult to measure incremental progress towards pattern machines that could be used for sequence transformation in robotics. Therefore, we introduce an adjustable-difficulty benchmark, where the transformations are procedurally generated using the probabilistic context-free grammar (PCFG) in Hupkes et al. [77]. These transformations include a collection of lexical rules that may be composed (e.g., reverse, shift, swap, repeat, etc.) over the tokens in the input sequence $x_{input}^i$ to generate $x_{output}^i$. Example transformations are given in Table 4 in the Appendix. The complexity

| Method | Accuracy (%) |
|---|---|
| (d3) text-davinci-003 | **75** |
| (d3) w/ random $\mathcal{A}$ | [†]$58 \pm 1$ |
| (p) PaLM [55, 56] | 74 |
| (d2) text-davinci-002 [53] | 69 |
| (d1) text-davinci-001 [39] | 60 |
| (c1) text-curie-001 | 54 |
| (b1) text-babbage-001 | 50 |
| (a1) text-ada-001 | 39 |

[†]Numbers averaged across 5 randomly sampled alphabets.

Tab. 2: LLMs of varying sizes are capable of completing patterns procedurally generated with PCFG, averaged over a range of $k$ and $w$.

of these transformations can be controlled by varying the number of tokens $k$ used to express sequences $x^i = (s_1,...,s_k)$, and the number of lexical rules $w$ used to define the transformation. This is simply the identity function when $w=0$, and progressively appears more complex as $w \rightarrow \infty$. Table 2 aggregates PCFG pattern completion accuracy across different LLMs over sequence length $k = [1,2,4,8,16,32]$ and complexity $w = [0,1,3,7,15,31]$, each with 100 runs (see Appendix A.4 for ablations of $k,w$). This benchmark

provides a more unbiased evaluation of pattern reasoning capabilities in LLMs; PCFG completion accuracy improves with model scale, and correlates with ARC performance. We use PCFG for evaluation only (rather than for training [77, 78]) so that one can measure how pre-training regimes or modalities may improve general pattern capabilities across sequence transformations. We have released the PCFG benchmark.

## 5    Sequence Completion

**Completion of sinusoids.** We start with a simple example where LLMs extrapolate a function of the form $f(x) = a \cdot \sin(bx)$. As in Section 4, tokenization matters; we found it effective to discretize outputs among integers 0–100, as these integers are represented by single tokens in the tokenizers of the LLMs we tested.

Fig. 3 shows completions of the sine wave by text-davinci-003 over 11 trials given 3 and 5 periods as context, as well as average distance (computed by Dynamic Time Warping) of the generated predictions to the ground truth function values across several LLMs. Multiple LLMs produce near-perfect continuations of the sine wave, especially with more context (i.e., more periods of the sine wave). We additionally test the function family $ax \cdot \sin(bx)$—in which the amplitude of the oscillations increases with $x$-values. Here, the LLM must extrapolate to new values unseen in the context, which highlights the utility of using a metric space for the outputs (0–100) where the LLM has priors over the scale of the different tokens. These functions also contain a "meta-pattern": the $y$-values increase, decrease, and then increase in a single period—and the amplitude of the function

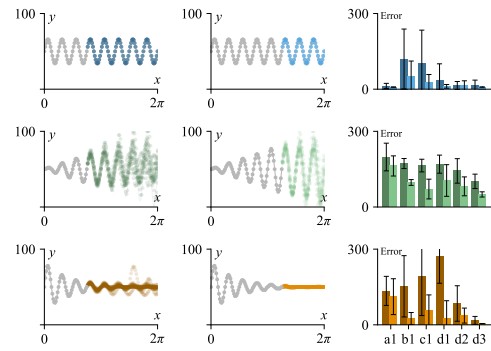

Fig. 3: LLMs (d3 shown) can extrapolate various functions $y = a \cdot \sin(bx)$ (top row), $y = ax \cdot \sin(bx)$ (middle row), and $y = \frac{a}{2^x} \sin(bx)$ (bottom row) given amounts of context. Overall, larger models make better predictions with lower error rates (right column). More context also helps prediction accuracy (light vs. dark).

also increases over time. This is a form of least-to-most prompting [69], an ability we find useful later for sequence improvement in Section 6. We also test the function $\frac{a}{2^x} \cdot \sin(bx)$. Across these three functions, we observe that greater context and larger scale LLMs yield higher quality predictions.

**Completion of periodic motions.** We emphasize that the Sequence Completion capability above is domain-agnostic—i.e., we do not use any specialized prompts explaining what function should be completed, nor do we provide any linguistic grounding for the metric tokens. We can therefore operationalize this zero-shot capability of LLMs to simple open-loop motion extrapolation problems in robotics, e.g., by encoding a series of positions sampled from a demonstration, and predicting future positions. We test two simple tasks on a mobile robot manipulator: *Table Sweeping* and *Whiteboard Drawing* (both shown in Fig. 2).

In *Table Sweeping*, the goal is to continue a human-provided kinesthetic demonstration of sweeping a portion of a table (see middle Fig. 2). We encode the demonstration as a series of end-effector poses at approximately 3 Hz. Each demonstration lasts roughly 20-30 seconds. We represent the 7-dim end-effector pose as a concatenation of Cartesian position and the quaternion, where each value is binned to an integer between 0 and 100, and the dimensions are delimited by spaces. We collect 30 demonstrations that demonstrate the sweeping motion. Note that demonstrations in this task are noisier and higher dimensional than the stylized sinusoid functions above. For each demonstration, we construct a context to consist of the first two-thirds of the provided demonstration, and treat the last one-third as the ground truth for the LLM to predict. Larger models quantitatively perform better with generally lower variance (see Appendix).

In *Whiteboard Drawing*, the goal is to continue a scripted demonstration of drawing loops on a whiteboard (see Fig. 2). Loops are defined by parametric equations of the form $x = a_x \cos(bt) + d_x$ and $y = a_y \sin(bt) + c_y t + d_y$. We execute the motions using position control and record the end-effector positions at 5 Hz, then discretize states in between 0 and 300, as finer motion is needed for this task. We provide part of the loop pattern in-context, and assess the ability to extrapolate from 2 loops to do a third loop. LLMs, e.g., text-davinci-003 perform well—we show more completions with different loop styles in the Appendix.

# 6 Sequence Improvement

In this section, we explore the synergies between sequence transformation and completion— and investigate *improving* a sequence, such as trajectories in a sequential decision process, along some metric, such as a reward function. Here, we use an LLM to generate new sequences $x^N$ conditioned on previous sequences $(x^1, ..., x^{N-1})$, which can represent previous iterations of the same sequence (or policy it represents). The improvement can also be return-conditioned, given a reward function $r(\cdot)$. By inserting as the first token(s) of each sequence its corresponding total reward $x = (r(x), s_1, ..., s_k)$, we can prompt the model to conditionally "improve" by "just asking" [79] for a higher reward than those seen in-context (i.e., prompting LLMs to act as Decision Transformers [80]). New "rollouts" can yield new reward labels that then replace the original desired rewards with actual rewards. Iteratively performing this inference and accumulating trajectories may jointly use the model's general notion of pattern transformation and extrapolation to perform improvement of sequences, which can be represented by numeric or symbolic tokens. Note that there are practical considerations, e.g., depending on the task or model, not all sequences can fit in context, so options could be to keep the most recent, or the ones with the highest rewards if available (see Appendix for more discussion). In this section, we perform a series of targeted experiments on simple tasks, aiming to explore the possibility of using LLMs for sequence improvement in trajectory and policy optimization.

**Extrapolating simple meta-patterns among trajectories.** Sequence improvement with LLMs enables a simple form of trajectory optimization for a *Marker in Cup* task on a Franka Panda robot, where we define the prefixed reward of a trajectory to be the negative distance between the final end-effector position and the cup (normalized between 0–100), and initialize the context with a collection of trajectories (stopping at 20%, 40%, 60%, and 80% of the way to the cup), delimited by newlines and prefixed by rewards (ranging roughly from 70-90; see Appendix). For this task, we represent trajectories as sequences of Cartesian positions, each dimension normalized between 0–100. We find that text-davinci-003, to an extent,

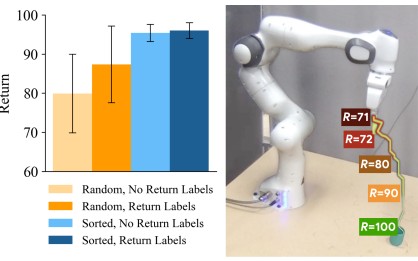

Fig. 4: LLM agents can generate new trajectories with increasing returns for a *Marker in Cup* task (right). Performance varies with different ways of building the context (left).

is able to generalize the pattern and generate a trajectory that achieves a reward $> 90$. For this extrapolation to occur, we observe that meta-patterns in the context are crucial: in Fig. 4 (left), we compare the average reward achieved by text-davinci-003 over 11 trials (each with a different goal position) given contexts with different trajectory orderings (sorted by reward, randomly permuted, or with/without reward annotations).

**Sampling higher-reward trajectories online.** While LLMs can extrapolate from trajectories that exhibit clear meta-patterns among them, we find that this ability is more limited for less trivial setups. Consider a simple $9 \times 9$ *Grid* navigation environment with a random goal position and a fixed starting position at the grid center. Episodes terminate after 20 timesteps, and the return is based on the distance from the agent to the goal at the final time step. This environment is inspired by the Dark Room environment from [62] but with a continuous reward function, reducing the exploration challenge. The agent may take actions (1-5) corresponding to moving right, up, left, down, and no-op. We initialize the context buffer with 20 trajectories of agent grid positions generated by a random policy, sorted by total cumulative rewards. These trajectories exhibit a more complicated meta-pattern than in the *Marker in Cup* task; we

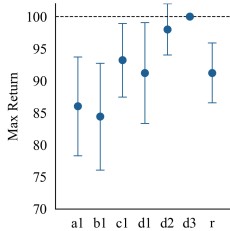

Fig. 5: Average max return for LLM agents a1-d3 on *Grid* compared to random exploration (r).

do not find that LLMs can generate trajectories of higher reward immediately. With that said, we can consider an iterative, *online* setting, in which the LLM acts as an agent that interacts with the environment in a closed-loop. The context consists of the highest reward trajectories in sorted order, appended with a higher reward than was seen in the context, plus states and actions from the current partial trajectory (see Appendix). Once an episode terminates, its trajectory is relabeled with the reward achieved, and inserted into the context at the appropriate position. In Fig. 5, we plot the maximum return attained by a1-d3 over 50 episodes, compared to random exploration, averaged over 5 trials. We find that a1-d1 tend to sometimes "exploit" the suboptimal behaviors represented in the context (which initially contains trajectories with rewards ranging from 6-78), whereas d3 can consistently find a solution to *Grid* within 50 episodes.

**Discovering a simple *CartPole* controller.** We show that using LLMs as agents in an online, closed-loop setting can discover a simple controller for *CartPole* (where observations consist of pole angle and velocity, normalized to 0–100, actions are 1 (left) and 2 (right), maximum horizon is 200). Fig. 6 (left) shows that return (number of steps the CartPole is kept upright) improves on average across various LLMs over 100 episodes (where the first 100 are generated by random exploration). Fig. 6 (right) shows the evolution of trajectories over episodes of d3, demonstrating that it discovers oscillatory behaviors to keep the CartPole upright.

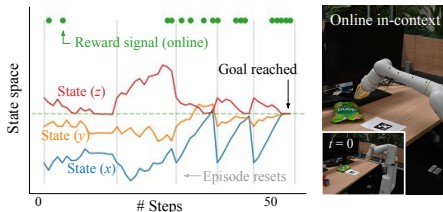

Fig. 6: Different LLM agents (d3 - c1) on average can improve trajectories (total rewards) with more *CartPole* episodes (left), and discovers "oscillatory behaviors" (right) to keep the CartPole upright (later episodes are brighter).

**Online human-guided trajectory optimization.** LLMs can also react to sparse binary reward signals (e.g., subjectively provided by a human) to adjust trajectories online. This is analogous to an implementation of "clicker training" [81, 82] used for training dogs, but instead applied to robots. In this setup, at every time step (2s), the robot executes an action corresponding to a movement of its end-effector in a particular direction. The human observes the action and chooses whether to give a

Fig. 7: LLMs can in-context react to sparse reward signals online to encourage an end effector to reach a desired goal.

reward (i.e., by using the clicker) to encourage or discourage similar behaviors. Episodes reset after 30 seconds, and the first two episodes are generated by random exploration. The (*reward*, *state*, *action*) tuples are added as in-context examples (with negative examples followed by positives, and an equal number of each) to generate the next action based on the current state. An example context format is given in the Appendix. As shown in Fig. 7, applying LLMs' sequence improvement capabilities in this way enables a human to guide the robot to push an object.

## 7    Discussion

We are excited about the opportunities of LLMs as pattern machines for robotics—from reasoning and extrapolating complex patterns as a prior for control, to online optimization of closed-loop policies via sequence improvement. These capabilities present several implications, including (i) perspectives on the role of language pre-training for end-to-end robot learning models [31, 32], and (ii) in-context learning of arbitrary patterns as a driving mechanism for policy improvement. LLMs also show promise for mixed autonomy settings—e.g., real-time pattern extrapolation for assistive teleoperation. We expect many of these abilities to continue improving as large models expand from learning patterns within language-only datasets to multimodal domains (e.g., images, videos). While this work investigates in-context generalization on fairly simple settings without additional data collection or model training, these capabilities presumably may be significantly improved via domain-specific objectives and finetuning [83, 84, 64, 65, 42].

**Limitations & Future Work.** Today, the inference costs (and monetary costs) of using LLMs in the control loop are quite high. Predicting the next token for every sequence, e.g., every dimension of every time step in a trajectory, involves querying an LLM. State-action spaces which are higher dimensional and/or greater precision also result in longer representations, and thereby the extent to which they can be extrapolated or sequence optimized is bounded by the context length of models. These limitations may prevent deploying these models on more complex tasks in practice; however, they may be partially mitigated by incorporating mechanisms like external memory, and by current efforts to drive improvements in LLM quantization [85] and inference efficiency [86]. An additional limitation lies in the fact that, for best performance, some care must be taken to represent patterns with consistent tokenization (which requires knowledge of the model's tokenization scheme). Finally, as with any other language-only model, LLM-based control may (i) be unpredictable, and (ii) lack visual/physical grounding; thus, it is not currently suitable for application outside of constrained lab settings. We leave the exploration of these important topics for future work.

## Acknowledgments

The authors would like to acknowledge Jie Tan, Peng Xu, Carolina Parada, Alexander Herzog, Jensen Gao, Joey Hejna, and Megha Srivastava for valuable feedback and discussions.

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

# A   Sequence Transformation

## A.1   Abstraction and Reasoning Corpus: Additional Details and Examples

In Section 4 of the main paper, we describe how ARC problems require reasoning about a range of different types of pattern operations—infilling, counting, translating and rotating shapes, and more. In Fig. 8, we show sample problems among the 800 ARC problems for which text-davinci-003 correctly generalizes the pattern shown in a few train examples to a test example. In Fig. 9, we show sample problems that are not correctly solved by text-davinci-003. In Listing 1, we show an example context for an ARC problem encoded as integers.

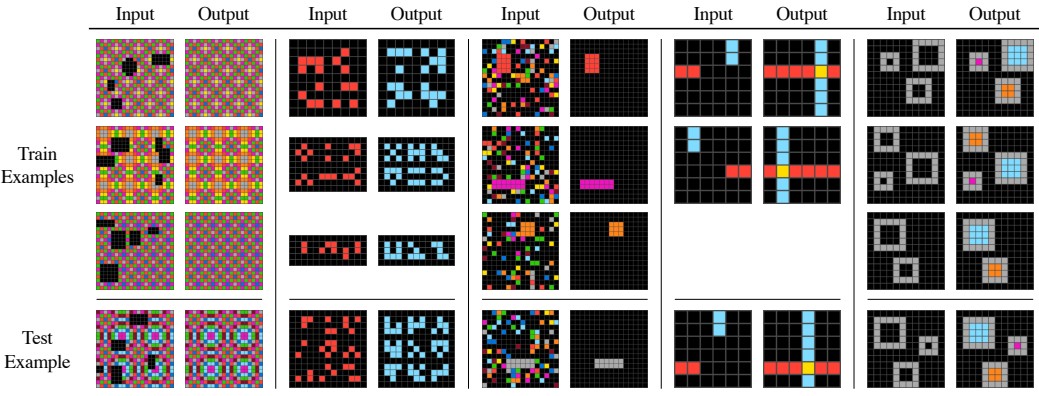

Fig. 8: Sample ARC problems that are correctly solved by text-davinci-003.

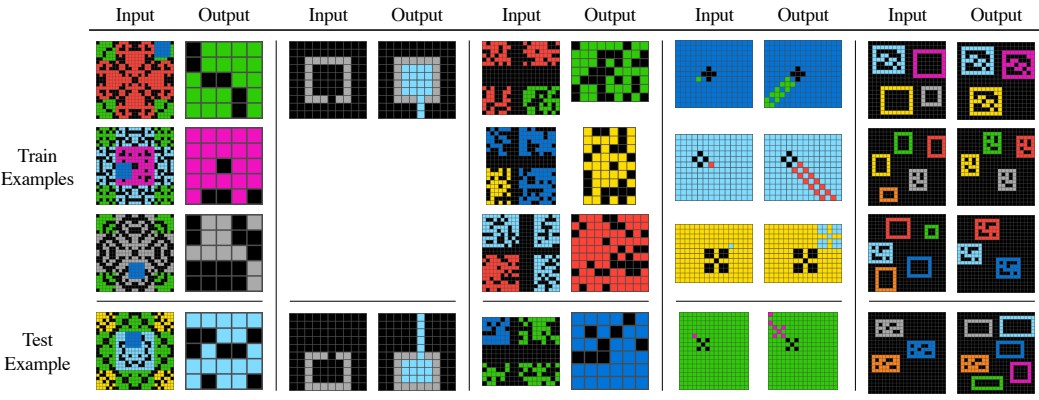

Fig. 9: Sample ARC problems that are not correctly solved by text-davinci-003.

```
input:
0, 0, 0, 0
0, 3, 4, 0
0, 7, 6, 0
0, 0, 0, 0
output:
3, 0, 0, 4
0, 0, 0, 4
0, 0, 0, 0
0, 0, 0, 0
7, 0, 0, 6
---
input:
0, 0, 0, 0
0, 5, 6, 0
```

```
0, 8, 3, 0
0, 0, 0, 0
output:
```

Listing 1: Example context format for an ARC problem (only one input-output example is shown, along with a query input.

## A.2 Patterns over Low-Resolution Images

In Fig. 10, we show an example in-context grasp detector which outputs target coordinates in a downsampled image, given 6 in-context examples, as well as an example of a simple forward dynamics model predicting spatial rearrangement of a red bowl into a green plate, given 9 in-context examples. As LLMs progress on the benchmarks discussed in Section 4, they may become more robust and precise at performing such tasks.

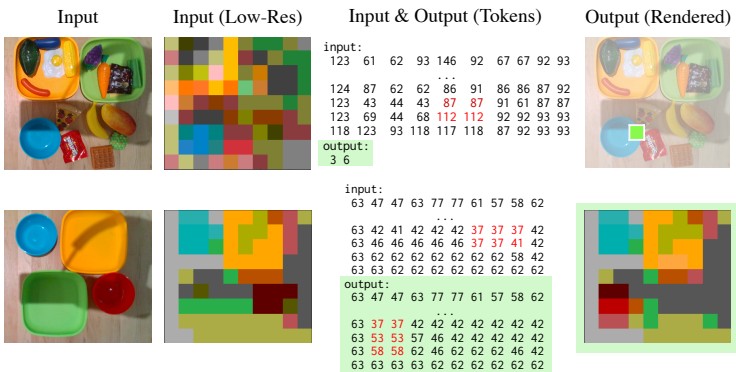

Fig. 10: Example LLM prediction as an in-context grasp detector (top) and a simple forward dynamics model (bottom).

## A.3 Token Invariance for New Token Embeddings

In Section 4, we have argued that LLMs are, to a certain extent, invariant to the choice of alphabet a pattern is encoded with, in line with prior work on mappings from semantically meaningful tokens to random tokens in a pre-trained language model [29, 30, 87]. Here, we present an experiment that investigates token invariance even further by introducing new token embedding vectors the model has not seen during training.

We sample $K$ many new embedding vectors as Gaussian samples using the mean and 1 or 2 standard deviations of the original LLM embedding matrix statistics in each of embedding vector dimension. This way, we create a new token embedding matrix, that mainly consists of the newly sampled token embeddings the model has not seen during training. Additionally, we add embeddings from the original matrix that correspond to separating tokens (comma, period) to build the input prompts. Although the model has never seen the new embedding vectors during training, we can feed them into the transformer as input and compute cosine similarities at the output analogously to how the original embedding matrix is treated.

Fig. 11 shows the success rate of correctly choosing the target token in a single-token prediction task when using the newly sampled embeddings in comparison with the native embedding matrix. The tasks we are considering are of the form (1, 1, 2) ↦ 2 or (1, 2, 2) ↦ 1. We provide in-context examples to build a prompt of the form "1, 2, 2, 1 \n 3, 4, 4, 3 \n 5, 6, 6, 5 \n 7, 8, 8," where the correct prediction should be "7" in this example. Note that the numbers "1", "2" etc. are randomly mapped to the newly sampled token embeddings for indexing purposes and in particular do not enter the LLM. As one can see in Fig. 11, for $1\sigma$ noise sampling, the model is able to solve the task with the new embeddings with similar performance as with the native embeddings. In case of $2\sigma$, the performance degrades. Although these are relatively simple single-token prediction tasks, this experiment shows that LLMs show pattern recognition abilities even when prompted with out-of-distribution continuous input embeddings. The results are obtained with $K = 100$, averaged over 3 random seeds when sampling the token embeddings, 30 instances each, and a context length of 5, 10, or 20 examples. The LLM is the 8B parameter variant of [55].

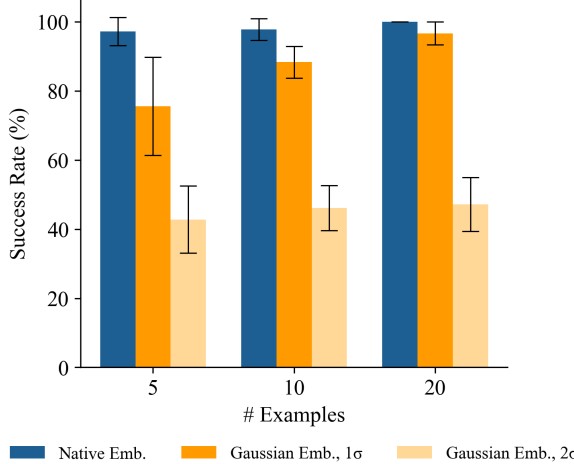

Fig. 11: Token-invariance experiment with newly sampled token embeddings the model has not seen during training. Shown are success rates when using randomly sampled token embeddings from the native embedding matrix, or newly sampled embeddings.

## A.4 PCFG Benchmark: Additional Details and Ablations

Our PCFG benchmark is a procedurally generated, adjustable-difficulty benchmark for measuring abstract sequence transformation capabilities in LLMs, based on the PCFG from [77]. In Table 3, we show illustrations of the primitive operations in the PCFG that can be applied on one or two sequences of tokens. In Table 4, we show examples of two transformations (of different complexities) from our benchmark, which are composed of the primitive operations. In Table 5, we show independent ablations of sequence length (number of tokens) $k$ and complexity (number of rules) $w$ in the sequence transformations, illustrating the way in which the solve rate decreases as either factor increases. In Listing 2, we show an example context for a PCFG problem on integer sequences.

| Unary Functions | | | Binary Functions | | |
|---|---|---|---|---|---|
| Name | Input | Output | Name | Input | Output |
| copy | ⬜🟧🟩 | ⬜🟧🟩 | append | ⬜🟧🟦🟪 | ⬜🟧🟦🟪 |
| reverse | ⬜🟧🟩 | 🟩🟧⬜ | prepend | ⬜🟧🟦🟪 | 🟦🟪⬜🟧 |
| shift | ⬜🟧🟩 | 🟩⬜🟧 | remove_first | ⬜🟧🟦🟪 | 🟦🟪 |
| swap | ⬜🟧🟩 | 🟩🟧⬜ | remove_second | ⬜🟧🟦🟪 | ⬜🟧 |
| repeat | ⬜🟧🟩 | ⬜🟧🟩⬜🟧🟩 | | | |
| echo | ⬜🟧🟩 | ⬜🟧🟩🟩 | | | |

Tab. 3: Illustrations depicting the unary and binary operators from Hupkes et al. 2020, which we use for our PCFG benchmark.

```
6 7 7 8 1 5 9 8 9, 1 5 9 8 9 7 7 6 6; 4 3 0 3 5 0 2 3 8; 5 0 2 3 8 3 3 4
    4; 1 3 3 3 7 0 1 9 9,
```

Listing 2: Example context format for a PCFG problem (two input-output examples are shown, along with a query input).

| Function | Example Inputs | Example Outputs |
|---|---|---|
| `remove_second(swap(`$s_1$`, `$s_2$`), `$s_3$`)` |  |  |
| `echo(copy(swap(swap(`
`prepend(remove_second(`
`swap(echo(`$s_1 s_2$`)), `$s_3 s_4$`), `$s_5 s_6 s_7 s_8 s_9 s_{10}$`)` |  |  |

Tab. 4: Illustrations of transformations in our PCFG benchmark. Row 1 shows a transformation composed of $w\!=\!2$ operations over $k\!=\!3$ tokens, and row 2 shows a transformation composed of $w\!=\!8$ operations over $k\!=\!10$ tokens, respectively. For each transformation function, we show two example inputs and the corresponding outputs.

| | text-davinci-003 | | | | | | | text-davinci-003 w/ random $\mathcal{A}$ | | | | | | | PaLM | | | | | |
|---|---|---|---|---|---|---|---|---|---|---|---|---|---|---|---|---|---|---|---|---|
| | | | $w$ | | | | | | | $w$ | | | | | | | $w$ | | | |
| $k$ | 0 | 1 | 3 | 7 | 15 | 31 | $k$ | 0 | 1 | 3 | 7 | 15 | 31 | $k$ | 0 | 1 | 3 | 7 | 15 | 31 |
| 1 | 100 | - | - | - | - | - | 1 | 92 | - | - | - | - | - | 1 | 100 | - | - | - | - | - |
| 2 | 100 | 100 | - | - | - | - | 2 | 91 | 92 | - | - | - | - | 2 | 100 | 100 | - | - | - | - |
| 4 | 100 | 100 | 100 | - | - | - | 4 | 93 | 92 | 93 | - | - | - | 4 | 100 | 100 | 100 | - | - | - |
| 8 | 100 | 99 | 95 | 92 | - | - | 8 | 88 | 82 | 62 | 49 | - | - | 8 | 100 | 89 | 74 | 82 | - | - |
| 16 | 100 | 86 | 59 | 4 | 47 | - | 16 | 84 | 64 | 32 | 17 | 22 | - | 16 | 100 | 78 | 57 | 51 | 58 | - |
| 32 | 100 | 74 | 32 | 14 | 12 | 22 | 32 | 83 | 40 | 13 | 8 | 9 | 12 | 32 | 100 | 68 | 23 | 18 | 22 | 34 |

Tab. 5: Solve rate (%) for PCFG across number of tokens $k$ and number of rules $w$ for different models.

## A.5 PCFG Benchmark: Program Synthesis

We have run DreamCoder [88] on our PCFG benchmark to contextualize the hardness of the task, and present the results in Table 6. We ran DreamCoder with two different sets of initial primitives:

- *PCFG Ops.* In this version, we provide DreamCoder with an initial set of primitives that corresponds to the exact set of unary and binary functions (from [77]) that the PCFG benchmark is based on: `copy`, `reverse`, `shift`, `swap`, `repeat`, `echo`, `append`, `prepend`, `remove_first`, `remove_second`. We also include a slicing operator `slice`, `length`, and integers 1–10.

- *List Ops.* In this version, we provide DreamCoder with a set of list primitives: `length`, `empty`, `singleton`, `range`, `append`, `map`, `reduce`, `true`, `not`, `and`, `or`, `sort`, `add`, `negate`, `equal`, `reverse`, `index`, `filter`, `slice`. These primitives are not specially designed for PCFG and are based on those used in [88, 89].

In both cases, the provided primitives are sufficient to define the transformations in the PCFG benchmark. For each $(k,w)$ pair in Table 6, we train on 100 task instances and report the number of tasks which get solved (i.e. a correct program that satisfies the training examples is discovered). We ran DreamCoder for 4 iterations and use the default hyperparameters and timeout. As we would expect, the version of DreamCoder with "oracle" access to the PCFG operations performs well, in several cases matching or exceeding the performance of LLMs. This is especially true when the search problem is easier (i.e. when number of functions $w$ is smaller). The version of DreamCoder with access to list primitives is also able to solve many of the tasks with small values of $w$, but there is a sizeable dropoff as the complexity of the tasks increases. These results help to contextualize the difficulty of the PCFG benchmark when given access to different amounts of domain-specific information. We also note that we would expect brute-force search over the PCFG operators to eventually solve these tasks. Doubling the computation time budget for the version with oracle access to the PCFG operators leads to increased success rates ($k\!=\!8$, $w\!=\!3$) increases from 80→84; ($k\!=\!16$, $w\!=\!7$) increases from 33→41.

| DreamCoder w/ PCFG Ops. | | | | | | DreamCoder w/ List Ops. | | | | | |
| --- | --- | --- | --- | --- | --- | --- | --- | --- | --- | --- | --- |
| | $w$ | | | | | | $w$ | | | | |
| $k$ | 0 | 1 | 3 | 7 | 15 | 31 |
| 1 | 100 | - | - | - | - | - |
| 2 | 100 | 100 | - | - | - | - |
| 4 | 100 | 100 | 98 | - | - | - |
| 8 | 100 | 100 | 80 | 58 | - | - |
| 16 | 100 | 98 | 64 | 38 | 38 | - |
| 32 | 100 | 96 | 41 | 24 | 26 | 37 |

| $k$ | 0 | 1 | 3 | 7 | 15 | 31 |
| --- | --- | --- | --- | --- | --- | --- |
| 1 | 100 | - | - | - | - | - |
| 2 | 100 | 100 | - | - | - | - |
| 4 | 100 | 85 | 51 | - | - | - |
| 8 | 100 | 81 | 23 | 25 | - | - |
| 16 | 100 | 63 | 26 | 4 | 17 | - |
| 32 | 100 | 66 | 6 | 3 | 5 | 11 |

Tab. 6: Solve rate (%) for PCFG across number of tokens $k$ and number of rules $w$ for DreamCoder initialized with two different sets of primitives.

# B  Sequence Completion

## B.1  Sinusoids: Structure Learning Comparison

While the sinusoid extrapolation could be easily performed with standard regression techniques, we contextualize the task with another method that has no specific prior knowledge of the function being extrapolated. We include the structure learning baseline from [90], implemented with Gen [90]. This method uses a Gaussian Process with an inferred covariance kernel to model time series data. The covariance kernel is inferred using MCMC sampling over a PCFG of covariance functions (e.g. squared exponential, periodic). We run the algorithm for 100 iterations and sample from the resulting GP, as shown below in Fig. 12. The training data is generally fit with low error. However, the quality of the completion differs for the various functions; the sine wave is generally extrapolated well whereas the sinusoids yield high variance samples. Similar to the LLMs, greater context generally yields lower error completions. Note however that the outputs of the structure learning algorithm are high-variance by design, and there are multiple ways to utilize the outputs of the algorithm. We also note that [90] was tested on a larger set of functions than those we look at here. Though not the goal of our work, it would be interesting future work to evaluate how LLMs extrapolate patterns generated by a wider array of function classes. We also refer to [42] for an extensive comparison of language models to baselines on regression tasks when formulated with a natural language interface as well as a study on the effects of fine-tuning.

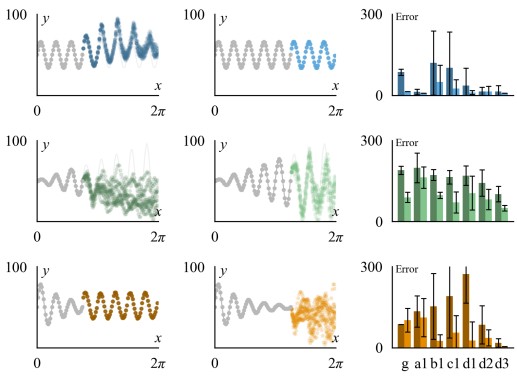

Fig. 12: The structure learning approach (g) extrapolates various functions $y = a \cdot \sin(bx)$ (top row), $y = ax \cdot \sin(bx)$ (middle row), and $y = \frac{a}{2^x} \sin(bx)$ (bottom row) with different degrees of error. More context also generally helps prediction accuracy (light vs. dark).

## B.2  Table Sweeping: Additional Details

In Section 5 of the main paper, we demonstrate how sequence completion capabilities can be applied to continuation of partial motions, such as sweeping a table. In Fig. 13, we show the average DTW distance between predicted and ground truth trajectory completions in the Table Sweeping task, given 66% of the trajectory as context, over 30 trials. Each full trajectory consists of 9 sweeping motions across a table. We compare completions made by various language models. We find that larger models generally

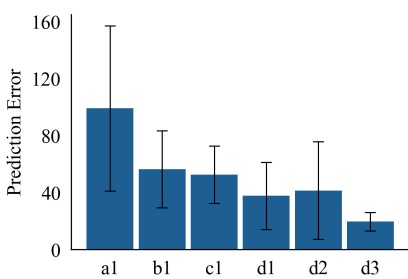

Fig. 13: LLM trajectory predictions *Table Sweeping* improve with larger models.

perform better; text-davinci-003 performs the best, and also has the lowest variance. On our website, we show qualitative examples of text-davinci-003 completing a table sweeping motion given by a human demonstration.

### B.3 Whiteboard Drawing: Qualitative Results

In Fig. 14, we show example completions for three different loop styles by text-davinci-003 over three trials. The completions generally match the overall shape shown in the two loops given as context. However, the results also qualitatively illustrate that fine motion patterns can be challenging to predict precisely.

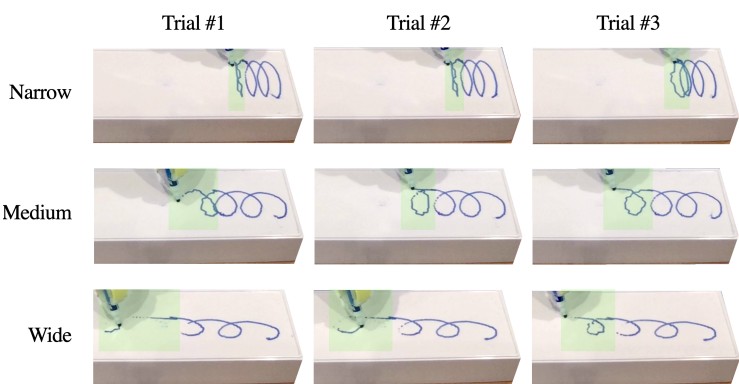

Fig. 14: Sampled drawings produced by performing an in-context completion (of one loop, highlighted in green) given a scripted demonstration of two loops. Each row is a different loop style (narrow, medium, wide), and each column is a different trial. Results are shown for text-davinci-003.

## C Sequence Improvement

### C.1 Marker in Cup: Additional Details

In this task, we use LLMs to generate improved trajectories (according to a reward metric) given a context of trajectories that have increasing returns. For this task, states are Cartesian $(x, y, z)$ positions, with each dimension normalized between 0 and 200, trajectories are series of states that can be executed via position control, and the return of a trajectory is proportional to the negative distance to the goal (cup) plus an offset. We form the trajectories in the context as follows: we take a full trajectory which attains a reward of 100 and construct trajectories that stop moving 20%, 40%, 60%, and 80% of the way to the goal (such that all trajectories are 50 timesteps). We condition the LLM to generate a 100-reward trajectory by prompting it with "100: *start state*". An excerpt of an example context is shown in Listing 3. The results in Figure 5 from the main paper are over 11 trials, each with a different goal position.

```
71: 104 83 123, 104 83 123, ...
72: 104 83 123, 104 83 123, ...
80: 104 83 123, 104 83 123, ...
```

```
90: 104 83 123, 104 83 123, 104 83 123, 104 83 123, 104 83 123, 104 83
    123, 104 83 123, 104 83 123, 104 83 123, 104 83 123, 104 83 123, 104
    83 123, 104 83 123, 104 83 123, 104 83 123, 105 83 123, 105 83 123,
    106 83 123, 106 83 123, 107 83 123, 108 83 122, 109 83 122, 110 83
    122, 111 83 121, 112 82 120, 113 82 119, 113 82 118, 114 81 118, 115
    81 117, 115 81 116, 115 80 115, 116 80 114, 116 80 113, 117 79 112,
    117 79 111, 118 79 110, 118 78 109, 118 78 109, 118 78 109, 118 78
    109, 118 78 109, 118 78 109, 118 78 109, 118 78 109, 118 78 109, 118
    78 109, 118 78 109, 118 78 109, 118 78 109, 118 78 109
100: 104 83 123
```
Listing 3: Example context (excerpt) for a Marker in Cup, illustrating the (*reward*: *state*, *state*, *state*...) format..

## C.2 Grid: Additional Details

In the *Grid* environment, observations are $x,y$ positions represented by integers 0–8 for each coordinate. There are five possible actions (1, 2, 3, 4, 5) corresponding to (right, up, left, down) movement by one space and no-op. A goal is randomly placed in the grid. The agent (which is initialized at the center position) receives a reward of 100 - 10 * distance from the goal to the agent's final position. Episodes terminate after 20 time steps. For our experiments, we limit the context length to 1024 tokens. At each iteration, the LLMs is prompted to generate a trajectory with the maximum seen return from the buffer plus a randomly selected offset of up to 20.

## C.3 CartPole: Additional Details

We use a simplified version of the CartPole enviornment in OpenAI Gym. Observations are two-dimensional (corresponding to pole angle and velocity, normalized to 0-100) and the maximum time horizon is 200. There are two possible actions (1, 2) corresponding to (left, right), and the agent gets +1 reward for every time step that the CartPole is kept upright. In Listing 4, we show an example context excerpt for CartPole, where a trajectory history is appended with an encoding of the current trajectory.

```
52: 40 50, 1, 40 54, 2, 41 49, 1, 41 54, 1, ...
60: 45 50, 2, 45 45, 1, 44 50, 2, 44 45, 1, ...
75: 52 50, 1, 52 55, 2, 53 50, 2, 53 46, 2, ...
98: 44 50, 1, 44 55, 2, 45 50,
```
Listing 4: Example context format for a CartPole run. A trajectory history (with each trajectory in the format *reward*: *observation*, *action*, *observation*, *action* ...) is followed by an encoding of the current trajectory, up to the current observation.

Below, we discuss some additional considerations for forming the context from the trajectory history.

*Context Length.* When context length is longer, more trajectories can fit in the context (which yields more in-context "training data" that could potentially be used to generalize to higher rewards, but also requires the LLM to attend over more tokens). Context length is a limiting factor of using current LLMs in our trajectory improvement setting: the number of tokens required to represent a trajectory history scales with the observation dimensionality, action dimensionality, time horizon, and number of trajectories. For our CartPole experiments, we limit the context to 1024 tokens (which is the maximum context length for text-ada-001, text-babbage-001, and text-curie-001 models).

*Action Representation.* In initial experiments, we found that the tokens used to represent the action space (e.g. "0" for left, "1" for right) can seemingly affect the ability of an LLM to improve trajectories in the online setting. For example, we observed that if "0" is included in the action space, LLMs may "default" to sampling "0" (likely due to token-specific priors). Therefore, for our experiments, we use 1-indexed integer action representations, which appears to alleviate the bias towards choosing a particular action. The fact that action representation can sometimes affect performance complements our observations in the Sequence Transformation section, in which we find that token mapping invariance holds to some extent, but not entirely.

## C.4 Clicker Training: Additional Details

In our clicker training example, the observation consists of the end-effector position and the approximate object position as determined by visual input, with the $(x,y,z)$ values normalized between 0 and 300.

Actions correspond to movements of the end-effector (normalized between 0 and 100, such that `50,50,50` represents no movement). A sample context is given in Listing 5.

```
0:  80,49,138,109,54,133;  45,44,55
0:  82,32,155,109,54,133;  48,59,48
0:  82,32,155,109,54,133;  48,59,48
1:  88,31,154,109,54,133;  45,54,43
1:  85,36,146,109,54,133;  57,54,46
1:  93,40,142,109,54,133;  44,52,43
1:  ...
```

Listing 5: Example context format for clicker training. (*Reward*, *observation*, *action*) tuples are ordered by reward (with a click corresponding to a reward of 1) with an equal number of reward 0 and reward 1 transitions represented in the context.

