# OpenReview forum: "Large Language Models as General Pattern Machines"
_robot-learning.org/CoRL/2023/Conference — CoRL 2023 Poster_

### Official Review · Reviewer_seqs · 2023-07-17

**Confidence:** 3
**Originality:** Good
**Technical Quality:** Good
**Clarity Of Presentation:** Very Good
**Impact:** 4

**Recommendation:**

Weak Accept: I recommend accepting the paper, but will not argue for my recommendation if the majority of other reviewers have a different opinion.

**Review:**

Strength

1.	The evaluation study on LLM’s ability to model general patterns is rich and convincing.
2.	This paper provides valuable engineer experience on LLM (e.g., the influence of the tokenizer) which I think is beneficial to many readers in various domains.
3.	The paper writing is well.

Weakness

1.	Experiments on robot tasks, like reaching targets or continuing the previous action patterns, are too simple and limited. Therefore, I feel that one of the paper’s main ideas, the opportunities to use LLMs as pattern machines for robotics, is not so persuasive.


**Quality Of The Limitations Section:**

Limitations are addressed clearly

**Questions For Rebuttal:**

1.	Why there is no evaluation for GPT-4? Given the much better reasoning ability of GPT-4 compared to text-davinci-003, more practical and realistic robotic tasks might be doable.

**Robotics Focus:**

Sufficient demonstration on hardware

**Summary Of Paper:**

This paper evaluates the LLM’s abilities to model general pattern sequences including transferring a sequence from a source format to a target format via in-context learning (sequence transformation), extrapolating a given sequence (sequence completion), and improving the sequence (in decision-making tasks) via in-context learning via up-side-down RL conditioned on rewards (Sequence Improvement). In addition, the paper shows that such abilities have the potential to transfer to robotic tasks.

**Summary Of Recommendation:**

In general, I think the evaluation study of LLM on modeling general patterns is insightful and valuable. But I feel the experiments in robotic tasks are too simple and limited, and don’t well support the storyline. I would like to give a borderline, but there is no such option. Given the soundness and the value of the evaluation study in the general patterns, I tend to accept this paper.

---

### Official Review · Reviewer_JNxR · 2023-07-17

**Confidence:** 4
**Originality:** Good
**Technical Quality:** Good
**Clarity Of Presentation:** Very Good
**Impact:** 2

**Recommendation:**

Weak Reject: I recommend rejecting the paper, but will not argue for my recommendation if the majority of other reviewers have a different opinion.

**Review:**

Summary of positives / generous interpretation of paper:

This paper stimulates an interesting conversation about the relationship between domain-specific pattern processing and generalist pattern processing in large ML models.  The focus on in-context learning keeps it fresh and distinct from the related and more developed topics of meta-learning and transfer learning.  The authors establish good breadth by defining three separate robotics-adjacent paradigms, and multiple environments per paradigm, in which to test the performance of several LLMs.  Writing and overall structure are clear and relatively easy to digest, and the Related Work section is appropriately scoped.  Decent attention to detail e.g. PCFG sequence complexity and probing which aspects of domain specification are crucial.  Limitations and assumptions present in the example domains are adequately highlighted and the authors remain sober in discussing the future prospects for leveraging LLMs in this way.  While it is not clear that the general pattern capability of LLMs maps onto frontier challenges in robot control and motion generation, this paper may help us think more precisely about the different kinds of pattern-based reasoning required for those challenges and where we should focus our efforts for developing appropriate generalist resources for them.


Summary of negatives / critical interpretation of paper:

As interesting as this paper is from a deep learning perspective, its relationship to robot control is rather superficial and the approach still requires significant hand-tailoring (discretization/tokenization, ranking, etc) that debatably isn't so distinct from efforts needed with domain-specific languages.  There is a sense in which what is shown is that it is possible to engineer strained analogies that enable solving basic problems in one domain with powerful, resource-hungry tools from another domain.  I think the interesting thing about this paper is that LLMs have general pattern manipulation capability (as the title conveys) that can be leveraged through ICL, and not that general pattern manipulation capability can solve carefully-defined low-dimensional position sequencing problems.
A roboticist needs to know what a general approach, e.g. search-based planning, is suitable and scalable for before they decide to formulate their application as such.  Likewise I think roboticists need to know whether it is useful or natural to formulate a control or motion generation problem as general pattern manipulation problem rather than to merely know that it is possible at a basic level.

Further comments on details of example problems:

* Sequence Transformation
    * The ~spatial operations in ARC and similarly the lexical rules of PCFG benchmarks are completely discrete (e.g., rotation only by exactly 90 degree intervals), aren't they?  If there is any sense in which these setups (such as recognizing that a raster of a 75-degree rotated image of a color wheel was "close" to the discretely 90-degree rotated raster of that color wheel), it would be interesting to know something about that, but I assume it is not the case and that "spatial" is a rather qualified term in this situation.
    * Flattening 2d grids to 1d sequences -- because the output is autoregressive, this would imply that model performance might substantially vary between one input and its transpose?  Part of me feels like these problems aren't "truly" 2d but I suppose tasks like infilling are something of a counterpoint.
    * I appreciate the sharing of details of how the tokenization matters, but I also would consider this fact to support my criticism that the hand-designed interface is a big part of the solution.
    * I also like the examination of token mapping invariance; it's a nice way to try to distill out the non-language component of what an LLM is doing.

* Sequence Completion / Sinusoids
    * Given the use of the 0-100 tokens which I am not surprised for LLMs to grasp as representing a continuous range, and the basic concept of neural nets from the simplest MLPs as "function approximators", it's hard to feel much from this section's results.
    * The function "reminiscent of a stabilizing controller" doesn't seem to add anything other than possibly inviting false assumptions from someone taking a quick look at the paper; there is no fighting of perturbations or opposing forces or dissipating energy here, it's just a pattern that becomes increasingly trivial with time.

* Sequence Completion on robot arm
    * Nit picks: the EE pose is a 7-dim value representing 6dof and 1 constraint, and it's not clear that there's much significant happening in any of the angular DoF.
    * The noisiness of the human demos is the main aspect of this problem that resembles the difference between robotics and abstract pattern games on symbol spaces, but as evidenced by the "partially successful example" video in the supplement it doesn't seem like anything more significant than averaging/smoothing is going on.

* Sequence improvement
    * I found this section to be the most interesting and the idea of in-context RL is compelling for getting a generalist system to solve a novel problem on the fly, including with human guidance.
    * The analysis of Fig 5 feels a little double-edged, though, by again revealing that nontrivial hand specifications are necessary to get the generalist agent to solve simpe problems in the novel domain.  The result feels like makeshift gradient ascent.
    * Cartpole -- For brevity I'll just say that the boolean action space makes this only the most rudimentary proof-of-concept.

Lastly, I very much appreciate the direct acknowledgment by the authors of the high costs of using LLMs for even these very pared-down applications.


**Quality Of The Limitations Section:**

Additional details required

**Questions For Rebuttal:**

It is hard to know what to request from the authors as I'm not sure one could get this approach to a stage relevant to roboicists without first passing through a good portion of the territory presented here (which I find less obviously relevant to roboticists).  The authors have also been largely forthcoming and transparent about the assumptions and limitations of their work, and though I could suggest some more caveats to incorporate, it would not address my core reservations.

I am nonetheless interested to hear the authors' response to my assessment and would pose the following as some key points to respond to:
1. The necessity of carefully designed tokenization/discretization, ranking schemes etc make it harder to distinguish this approach as "generalist" in a way that domain-specific languages are not.
2. The generalization shown on ~continuous systems looks more like steering (directly modulating open-loop behavior) than adapting (responding to novel stimulus in a way that is not trivial to pre-specify); I think the latter is much more key to advancing robotics.
3. I posit that language has specifically evolved to solve the more abstract levels of decision-making and pattern processing -- more like the "code as policies" level and above, that are _complementary_ to the sorts of high-dimensional spatiotemporal reactivity / fine motor skills that are still insufficient in so many robotics applications and which we as humans acquire primarily by nature and practice rather than symbolic reasoning and communication.  Therefore it is little surprise to me that casting such problems as general pattern processing appears to be a strained exercise.


**Robotics Focus:**

Relevant but unlikely to deploy to hardware in near future

**Summary Of Paper:**

The authors apply LLMs to non-language tasks via ICL (in-context learning; examples are kept in context buffer rather than used to update weights). They test this ability for general pattern manipulation in abstract tasks on grids of symbols (as "sequence transformation") and in hand-tailored discretizations of position sequence extrapolation in 1d, 2d, and 3d (as "sequence completion" and "sequence improvement").  Their highlighted implications include use of LLMs for control (as priors or for online improvement) and the broad possibility that pretraining on one data modality can provide pattern processing capabilities useful for other data modalities (with a nod to the prospect of pretraining in additional modalities apart from language).  They note primary limitations of the high resource demand of using LLMs relative to standard control approaches, potential scaling issues with context buffer size, and unpredictability and lack of visual grounding of LLM-based control.

**Summary Of Recommendation:**

As someone with a general interest in deep learning, I found the paper stimulating and do not regret having spent time with it.  However, as a robotics practitioner and robot learning researcher, I would hesitate to suggest to my collaborators within these topic areas that they spend time on this paper.  Its clear presentation and decent attention to caveats/limitations somewhat limit the risk of adding more noise to a research area that has an overwhelming pace of activity, and I don't necessarily think a new idea needs to have immediately competitive results in order to be worthy of attention.  However, the level of care that went into this paper seems out of proportion with its questionable meaningfulness and appropriateness for what it is aiming to solve.  I wouldn't want it to crowd out other submissions that more clearly relate to the core challenges of robotics and the frontiers of motion generation and control.

---

### Official Review · Reviewer_A4xe · 2023-07-20

**Confidence:** 3
**Originality:** Fair
**Technical Quality:** Fair
**Clarity Of Presentation:** Very Good
**Impact:** 2

**Recommendation:**

Weak Reject: I recommend rejecting the paper, but will not argue for my recommendation if the majority of other reviewers have a different opinion.

**Review:**

**Major comments:**

*Representing the state-of-the-art non-LLM approaches:* I am not quite sure what the source of the authors number for Alford 2022 [ref-23]. If I read correctly, the author tested on smaller subset of symmetry problems for ARC and achieved a high success rate with an algorithm based on the dreamcoder framework [1] that compresses primitive operations into macros. Also, the state-of-the-art Kaggle success rate at present is 175/400 and not 130/800 [2]. Therefore, I urge the authors to remove the claim that LLMs are the best generalist solutions to ARC challenge. Similarly, the authors do not compare to function extrapolation methods for the sequence extension task. The automated time series structure discovery [3 sec 7.2 and refs therein]  performs extrapolation for a various function classes detected automatically, and it is in some sense general purpose, though with a different notion of generalization than an LLM. The purpose once again is not a direct comparison of the approaches, but a more accurate representation of what the state-of-the-art performance is in each of the domains considered by the authors and testing the LLM on an appropriate diversity of input values before making generalization claims

*Impact of tokenization:* The tokenization and compression of each feature vector into an integer between 0-100 for the sequence extrapolation and improvement tasks, and consistent tokenization for the sequence transformation task appears to be key to obtaining reasonable performance. it would be interesting if the authors experimented with another set of non-numerical but ordinal tokens (a-z or A-Z) representing the trajectories, or by associating a non-linear grounding to the numbers. The grounding problem of transforming the LLM output DSL (simple as it may be) into robot states is still completely solved manually by the user.

*Latency in continuous control tasks:* While the use of LLMs might be interesting as an intellectual excercise, in all but quasistatic robotics tasks, the latency of token generation might not be enough to operationalize a robot controller. The cost of each token generated by an LLM is a lot larger than computational cost of even using a policy network. This along with the inability of hosting the models locally, makes them particularly unsuitable for robotics task as identified by the authors in the limitations section. I would like the authors to reflect more on why an LLM-based approach might have benefits in the first place?

*Structure of the paper:* Currently, the paper very much reads as a series of case-studies, and I wonder if a restructuring of the text in terms of methodologies, tests/hypothesis, results and discussion in each subproblem would serve the draft a lot better.

[1] - Ellis, K., Wong, L., Nye, M., Sable-Meyer, M., Cary, L., Anaya Pozo, L., Hewitt, L., Solar-Lezama, A. and Tenenbaum, J.B., 2023. DreamCoder: growing generalizable, interpretable knowledge with wake–sleep Bayesian program learning. Philosophical Transactions of the Royal Society A, 381(2251), p.20220050.
[2] - https://www.kaggle.com/competitions/abstraction-and-reasoning-challenge/discussion/154597
[3] - Cusumano-Towner, Marco F., Feras A. Saad, Alexander K. Lew, and Vikash K. Mansinghka. "Gen: a general-purpose probabilistic programming system with programmable inference." In Proceedings of the 40th acm sigplan conference on programming language design and implementation, pp. 221-236. 2019.

**Quality Of The Limitations Section:**

Additional details required

**Questions For Rebuttal:**

1) Could you please clarify the non-LLM state of the art performance on the first two sequence reasoning tasks? Again this does not invalidate the approach, but is important to establish the best performance on these benchmarks.

2) What are the input output encodings for the ARC benchmarks for non-LLM approaches, do they consider image inputs or is there a recommended input, output representation format.

3) Non-LLM approach on PCFG benchmark? Dreamcoder [1] would be a valid baseline

4) What was the motivation for the construction of the PCFG benchmark? I notice that the all the operations involve a reuse of the input symbols. There are other sequence reasoning benchmarks in prior work that also handle substitution operations [4]


[4] - Ellis, K., Solar-Lezama, A. and Tenenbaum, J., 2015. Unsupervised learning by program synthesis. Advances in neural information processing systems, 28.

**Robotics Focus:**

Relevant but unlikely to deploy to hardware in near future

**Summary Of Paper:**

Summary:
This paper examines the role of large language models as a pattern reasoner and tries to operationalize pattern reasoning capabilities of LLMs to different types of robotics tasks involving sequence reasoning, or working with serialized data (such as rasterized images).

The authors present three use cases of pattern reasoning: sequence transformation, sequence extrapolation, and sequence evolution where the LLM output is expected to improve a labeled score of the sequence.

The authors present each of these sequence reasoning problems as a case study with observed performance and some discussion of the relevance of the observed results.

**Summary Of Recommendation:**

My primary concerns with the paper is that this is an attempt to apply LLM methods to a well studied area of ML. In doing so, the paper must adequately establish baseline performance on the problems it is considering, and as described in prior section, the authors have not done an adequate job of establishing the state of the art.

To my knowledge, establishing the baseline performance does undermine the claim that LLMs are general pattern machines to some extent. However, the ideas explored in the paper are interesting by themselves, and a more in depth discussion of the non-LLM baselines will only strengthen the paper

---

### Official Review · Reviewer_fBNs · 2023-07-21

**Confidence:** 4
**Originality:** Very Good
**Technical Quality:** Very Good
**Clarity Of Presentation:** Very Good
**Impact:** 4

**Recommendation:**

Strong Accept: I recommend accepting the paper and will argue for my recommendation even if other reviewers hold a different opinion.

**Review:**

This is overall an an excellent paper. It investigates the potential of using LLM for robotics from a relatively refreshing perspective. It nicely blends robotics-specific concerns and general concerns about sequence handling and learning generalization, and thus provides a principled study in robot learning.

The perspective and methods are novel and impactful. A fairly clearly favorable comparison here is that with Gato -- a much larger engineering effort -- in terms of arriving at less intuitive results, raising interesting questions, generating follow-up studies, and possibly leading to eventual real-world use.

The paper is generally clearly written and the video clips on the website were quite helpful.

The main concern is with the inadequate grounding of the analytical framework, specifically in terms of (1) situating the study in prior literature about pattern generation (e.g. from neuroscience), (2) motivating the three-way distinction of "sequence transformation", "sequence completion", and "sequence improvement" through situating them in robotics, and (3) raising and at least acknowledging, if not also systematically investigating, the real conceptual and architectural issues with tokenization and sequence preparation in the context of robotics and embodiment, beyond just affirming their importance. This concern leads the reviewer to believe that the limitation section as is currently written is inadequate (see more details below under "Questions For Rebuttal"). Another related concern is that because of the inadequate articulation of a strong analytical framework, the reader gets the sense that the number of studies reported is more than necessary but penetration into the substantive issues for future studies is still limited.


**Quality Of The Limitations Section:**

Additional details required

**Questions For Rebuttal:**

### Typo

- Line 355: "many these" => "many of these"

### Substantive Issues

##### 1. "Related Work" section

1.1 It might help if a table could be included or at least some heads or conceptual points highlighted so that the *themes* that matter could be grokked by the reader.

1.1 Also, it will be helpful, e.g. for supporting a more adequate analytical framework (see above and below), if relevant studies beyond LLM but rather from neuroscience about pattern generation and from robotics about sequence handling could be discussed.

##### 2. "Language Models as General Pattern Machines" section

2.1 What are the rationales behind the three-way distinction of "sequence transformation, sequence completion, and sequence improvement"? Does it come from robotics practice? Is it a consequence of first generating a sequence and then execute a sequence? Or is it merely consequent of seq2seq? Note that without enough framing here, it is a bit hard for one to tell which experimental studies and how many of them are enough for making the case the paper wants to make.

2.2 Concretely, for example, how should one understand the importance of sorting (Figure 5 and line 303-305) the sequences? Is it something one should expect? Is it surprising? Is it acceptable to sort in practice? (See 3. below for more.)

##### 3. Grounding and Challenging the Method

3.1 Tokenization

3.1.1 While the paper specifically addresses the importance of tokenization, a general point could be made: much "intelligence" comes from choosing the right tokanization scheme suitable for the specific tasks and environments. (Claim of a "generalist" rests on the intelligence that went into choosing tokenization. This is a point not adequately addressed by Gato and the need for training there made it much less interesting than the current study.)

3.1.2 Case in point: the ASCII art encoding for ARC does a lot of work and it requires privileged knowledge of tokens the LLM depend on. The study here was done decently adequately by including random choice of tokens, but analysis of tokenization as a system dependency and acknowledgement of the associated limitation seem to be absent. (E.g. discussions in subsection starting on Line 169 is both a contribution and comes with a limitation.)

3.1.3 Another case in point: discussion in Lines 189-195 has very important implication for robotics architecture, concerning how to understand the interface between a "perception module" and a "planning module". What is being illustrated here seems to be that there is no need for a vocabulary with fixed semantics to serve at such an “interface”. Instead, so long as there is adequately token differentiation with in-context semantics/correspondence, that may be enough. While this could mean less human-oriented interpretability, it could provide better generalization.

3.1.4 Time and tokenization: the temporal dimension of tokenization (and discretization) also matters as is clear from discussions around
Line 260 (3Hz) and Line 270 (5Hz). This is a specific place where prior study in neuroscience on Central Pattern Generator (cf.  https://en.wikipedia.org/wiki/Central_pattern_generator) becomes relevant. If we are to develop towards an end-to-end, fully adaptive method, with something like an LLM in the middle serving as pattern generator, then issues with real time must be handled in a principled approach.

3.1.5 Sequence forming: sorting discussed in Lines 303-305 (also mentioned above in 2.2) should be considered an issue belonging in the same family as tokenization. Again, a stronger analytical framework may be helpful for raising the right questions here or at least pointing out the limitations here for future work.

3.2 Extra mechanisms: control flow and external memory required for running the sequence improvement either online or offline could ideally also be contextualized in a more adequate framework.



**Robotics Focus:**

Sufficient demonstration on hardware

**Summary Of Paper:**

The paper investigates the potential use of LLM in robotics, without fine-tuning or retraining, for pattern generation in a sequential context. Tokenization and sequence construction methods were developed for tasks, under three categories which the authors call "sequence transformation, sequence completion, and sequence improvement", including some pattern completion tasks from the Abstract Reasoning Corpus, the classical closed-loop control task CartPole, open-loop operations of robotic arms, and a few others. Systematic and rigorous studies of the effectiveness of the methods were carried out to conclusively demonstrate that there is real potential in using LLM for pattern generation in robotics.

**Summary Of Recommendation:**

Clearly an excellent paper, possibly an outstanding one for CoRL. It has its limitations in terms of the depth of the study, but one can only do so much in a first work in a novel direction. The work here is very likely going to be impactful and stimulate substantive follow-up studies.

---

### Author Response · Authors · 2023-08-12
**Response to All Reviewers**

We thank the reviewers for their thoughtful reviews. We appreciated the comments that our work is “systematic and rigorous” [Rev. fBNs], “insightful and valuable” [Rev. seqs] with “interesting ideas” [Rev. A4xe] and a “fresh and distinct” [Rev. JNxR] perspective. It is exciting that the reviewers found that our work “stimulates an interesting conversation” [Rev. JNxR] and “is very likely going to be impactful and stimulate substantive follow-up studies” [Rev. fBNs].

We have provided point-by-point responses to each of the reviewers as a comment under each individual review.

Additionally, we would like to briefly summarize our response to two points that are raised by multiple reviewers.

1. *Tokenization is a form of hand-tailoring and/or requires information about the model.*

While some knowledge about the tokenizer is required, this is generally not privileged information about the model and is readily available, so it is not a particularly strong assumption. We also respectfully disagree that our tokenization involves significant "hand-tailoring." As our experiments in Section 4 show, we only need consistent tokenization (mapping units of information in the pattern to single tokens), which is a generalizable and easy-to-implement idea. Furthermore, our results on random token mappings suggest that once the token vocabulary is known, LLMs can perform sequence transformations to some extent even with randomly-sampled mappings from units in the patterns to tokens. To further investigate this token mapping invariance, we have added an additional experiment focusing on continuous-valued embeddings, showing that some basic sequence transformation capability in LLMs is retained even for randomly sampled continuous embeddings that are not associated with any token in the vocabulary.

2. *LLMs are not the best tool for these tasks.*

We would like to clarify that the intention of our work is not to suggest that today’s LLMs outperform specialized methods, or that they should be used as the primary tool for the discussed experiments! Instead, our intention is to convey a more modest yet exciting message – that there exists evidence showcasing the ability of large pretrained models to manipulate non-linguistic patterns. While the immediate practical implications may be limited, this surprising evidence could significantly influence how we perceive the use of pretrained large models in robotics.
We speculate on the potential impact of this finding on future research in the following ways:
1) Expanding the use of existing LLMs beyond high-level reasoning for generating data (e.g. continuing a particular pattern, or optimizing for higher reward behaviors).
2) Recognizing the need for addressing the pressing practical limitations of large pretrained models, such as latency and context length.
3) Evaluating pretrained large models on how well they store transferable patterns that may be useful for robotics applications.

**We have attached a revised version of the paper + appendix to the individual responses, incorporating many comments from the reviewers.** We have marked portions with substantive changes in blue font. The appendix contains information about 3 new experiments: results of DreamCoder on PCFG (Appendix A.3), results of a structure learning algorithm on sequence completion (Appendix B.1), and results of new sequence transformation experiments involving continuous token embeddings (Appendix A.4).

---

### Author Response · Authors · 2023-08-16
**Note to Area Chair(s)**

For the Area Chair(s),

We greatly appreciate the reviewers’ thorough engagement with our work!

Thanks to good feedback and comments, we had an opportunity to improve the manuscript by adding additional clarifications in the writing, and running additional experiments according to reviewers' suggestions.

A couple of the remaining concerns are based on perspectives about the appropriateness of connecting LLMs to robotics in the ways we have examined, as well as the robotics audience, rather than concerns about the technical novelty, clarity, or rigor of our work.

> [Reviewer A4xe] Application oriented works with LLMs are still of great value to robotics. Here I echo JNxR that the robotics applications of this approach are still quite strained. However this is merely an "editorial" opinion. While I retain my score for now, I agree that the draft is now of a very high quality and would not oppose its acceptance on the grounds scholarship, soundness and novelty.

> [Reviewer JNxR] Its clear presentation and decent attention to caveats/limitations somewhat limit the risk of adding more noise to a research area that has an overwhelming pace of activity, and I don't necessarily think a new idea needs to have immediately competitive results in order to be worthy of attention. However, the level of care that went into this paper seems out of proportion with its questionable meaningfulness and appropriateness for what it is aiming to solve.

> [Reviewer JNxR] I remain in agreement that the paper is quite interesting for [an ML context], where generalization of LLMs at this level of decision-making is not widely discussed. The paper is probably developed enough to be a strong positive contribution there [...] I also remain skeptical that the ties and gestures to [a robotics context] are as of yet meaningful enough to be more driving than distracting in that community (i.e. CoRL).

Regarding appropriateness of the approach:

We'd like to highlight that part of the motivation of our work is precisely to question the predominant intuition that large pretrained models – current and future – are only useful at language levels of abstraction for robotics. We hope that our observations may help spotlight a few of their non-linguistic capabilities too, and encourage initial (and future) discussions that may give rise to a better understanding of how they may be useful in robotics beyond high-level reasoning – not to suggest that LLMs today are anywhere close to replacing specialized algorithms.

Regarding the venue:

We acknowledge that the non-linguistic tasks we investigate are simple when viewed purely from a robotics perspective, and so it may seem that our contributions are more relevant to a general ML community. However, we believe that robotics is one domain that may stand to benefit substantially from LLM’s non-linguistic capabilities. In highlighting the connection to problems in robotics, we hope that works like this one may help influence evaluations for the future development of large scale models in the ML community at large. We think it would be valuable to bring some of the topics explored in both communities closer together.

We hope that the Chair would be willing to consider these points during the discussions and decision-making – we are open to suggestions, and more than happy to change how the work is presented wherever needed.

---

### Decision · Program_Chairs · 2023-08-30

**Decision:**

Accept (Poster)

**Comment:**

The paper investigates the ability of LLMs to recognize and manipulate general patterns within in-context learning with abstract sequences, divides the problem into three subcategories, and shows first transfers to robotic domains.
While in general the reviewers tend to agree on the novelty and interestingness of the idea – beside the clear presentation and appreciated thoroughly executed studies – there are mixed opinions on the applicability to robotics and the investigated problems with respect to the (robotic) claims. Some main concerns raised tackle the positioning of the study in prior literature, the task complexity in the evaluations, and the ‘general utility’ of the approach (speaking simplified and overexaggerated).
Nevertheless, the paper shows a refreshing approach and thinking, that may spark many interesting follow up research questions, and quite often such novel ideas produce split opinions at first. The authors provided an extensive update during the rebuttal and there was a nice and intensive discussion.
While I share some of the concerns of the reviewers with respect to the robotics/CoRL relevance, I also see the value of discussing refreshing and novel ideas in a different context and feel the novel connection of LLMs and robotics – in combination with the high quality of the paper – is of great value to the CoRL audience.

**Importantly though, the updated paper is a full page too long. The authors must shorten the paper significantly without removing important and significant content!**